# Auto-ACD: A Large-scale Dataset for Audio-Language Representation Learning

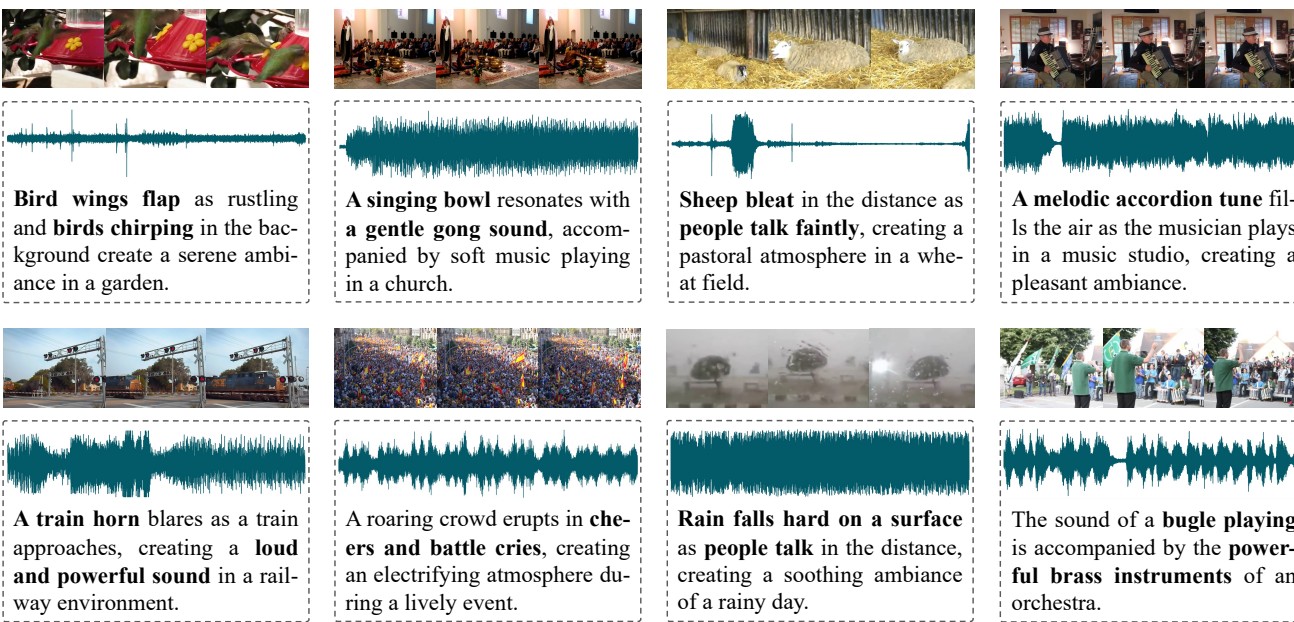

**Figure 1: Eight examples from the proposed Auto-ACD. It is a large-scale audio-language dataset with massive audio-text pairs (1.5M), long sentences (18 words) and diverse vocabularies (23K), consisting of audio separated from videos and captions generated by an automatic pipeline. Auto-ACD comprises more elaborate sound descriptions, more abundant auditory incidents and unique environmental information. The pivotal sound events are highlighted in bold.**

## ABSTRACT

Recently, the AI community has made significant strides in developing powerful foundation models, driven by large-scale multimodal datasets. However, for audio representation learning, the present datasets suffer from limitations in the following aspects: insufficient volume, simplistic content, and arduous collection procedures. To establish an audio dataset with high-quality captions, we propose an innovative, automatic approach leveraging multimodal inputs, such as video frames, audio streams. Specifically, we construct a *large-scale, high-quality, audio-language dataset*, named as **Auto-ACD**, comprising over 1.5M audio-text pairs. We exploit a series of pre-trained models or APIs, to determine audio-visual synchronisation, generate image captions, object detection, or audio tags for specific videos. Subsequently, we employ LLM to paraphrase a congruent caption for each audio, guided by the extracted multi-modality clues. To demonstrate the effectiveness of the proposed dataset, we train widely used models on our dataset and show performance improvement on various downstream tasks, namely, audio-language retrieval, audio captioning, zero-shot classification. In addition, we establish a novel benchmark with environmental information and provide a benchmark for audio-text tasks.

## CCS CONCEPTS

• **Computing methodologies** → **Computer vision**; • **Information systems** → **Data structures**; **Information retrieval**.

## KEYWORDS

audio-language dataset, audio-language contrastive learning, audio-visual correspondence, automated audio captioning

**ACM Reference Format:**
Anonymous Author(s). 2024. Auto-ACD: A Large-scale Dataset for Audio-Language Representation Learning. In *Proceedings of ACM Multimedia (ACM MM)*. ACM, New York, NY, USA, 10 pages. https://doi.org/XXXXXXX.XXXXXXX

# 1 INTRODUCTION

In recent literature, foundation models, like CLIP [45], variants of GPT [46], DALL-E 2 [47] and Stable Diffusion [49], have shown tremendous success in various understanding and generation tasks. Despite being different in architectural or algorithmic designs, they are fundamentally lying on a common basis: large-scale multimodal datasets, for example, MMC4 [59], LAION [50], HowTo100M [36], indicating an emerging transition from a model-centric to data-centric representation learning. The former considers pushing the boundaries of model design within the constraints of a predetermined data budget, while the latter focuses on curating large-scale and high-quality datasets in a scalable manner.

In the audio community, there have been recent endeavours focused on constructing audio-language datasets, as demonstrated in Fig. 2. However, existing datasets potentially suffer from two limitations, laborious and complicated collection processes and simplistic descriptions in text. On the one hand, Clotho [10] and AudioCaps [21], which contain audios typically comprising 1 to 3 sound events, accompanied by high-quality text descriptions provided by human annotators. They are clearly challenging to scale up. On the other hand, LAION-Audio-630K [53] and WavCaps [35] source large amounts of raw data from online foley websites, then employ sentence templates or keyword-to-caption models to convert the original audio labels into free-form sentences. However, it is obvious that the resulting language descriptions indeed offer little additional information beyond simple prompts or sound tags. Models trained on these datasets are incapable of learning robust audio-language representations. Furthermore, an exemplary audio caption ought to encapsulate four varieties of information: the 'what' - the nature of the sound perceived, the 'who' - the entity producing the sound, the 'how' - the characteristics of the sound, and the 'where' - the location the sound occurs.

This paper presents our recent efforts for constructing a large-scale, high-quality, audio-language dataset, with minimal manual efforts, termed **Auto-ACD** (**A**udio **C**aptioning **D**ataset by **Auto**matic Collection), with massive audio-text pairs (1.5M), long texts (18 words) and diverse vocabularies (23K). Our key insight is that humans do not solely rely on audio to understand audio accurately. A comprehensive understanding of the visual scene is expected to serve as a valuable information source and is sometimes necessary for understanding the audio content. Therefore, we build Auto-ACD on the prior of robust audio-visual correspondence in existing audio-visual datasets, for example, VGGSound [6], AudioSet [12]. Particularly, we employ a range of publicly available tools or APIs across the general AI community, *e.g.*, vision, language and audio models, to generate comprehensive language descriptions for the audio of the given video datasets. As a result, such descriptions not only depict the type of sound and its source, but also describe the auditory attributes and the specific location of its occurrence. Due to the limited information in audio tags, these pieces of information are infrequently present within the existing datasets.

The factual robustness in audio-visual correspondence significantly surpasses the capabilities of AI tools or APIs. Distinct from approaches that employ the 'teacher-student' model for data synthesis to augment training, we enrich audio captioning generation with information from an additional modality. Furthermore, we

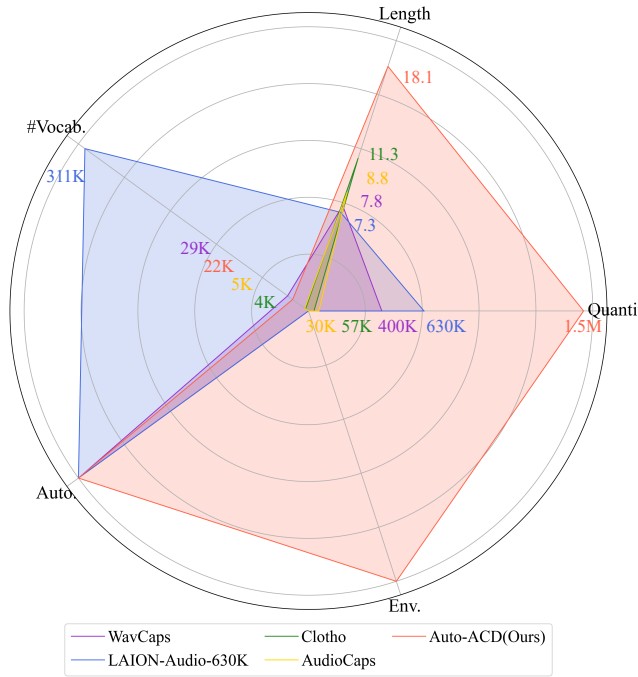

Figure 2: Comparison with other audio caption datasets. "Length" and "# Vocab." refer to average length and vocabulary. "Env." and "Auto." refer to environmental information and automatic pipeline, respectively.

utilize multiple cross-modality models to generate a variety of multi-modality inputs. We then employ a large language model (LLM) to collectively assimilate all inputs, identify and eliminate any illogical information, and generate comprehensive descriptions for the audio. These rich and complementary multi-modality inputs offer extensive guidance information that surpasses what audio captioning requires. Models trained on our dataset will transcend the limitations typically associated with AI tools and learn more robust audio-language representations.

To comprehensively validate auditory representation, for instance, audio events, and ambient information, learned from the text descriptions of Auto-ACD, we conduct experiments from four perspectives: *First*, we launch a joint audio-language representation learning using InfoNCE loss [15, 42], and evaluate the model through a retrieval task between audio and language, showing noticeable improvement over existing datasets; *Second*, we conduct zero-shot classification experiments, thus substantiating the accurate environmental information encapsulated within our dataset; *Third*, we benchmark on audio-language generation task, specifically, automatic audio captioning, by training a lightweight mapping network between the pre-trained audio backbone and GPT2 [46], showing superior performance on the widely used benchmark, *e.g.*, Clotho [10]; *Fourth*, we manually filter a test set and introduce a novel benchmark for audio-text tasks. This benchmark assesses the ability of models to grasp information beyond mere audio tags, for example, the environment and fine-grained categories of sound, we set a baseline for future research in this field.

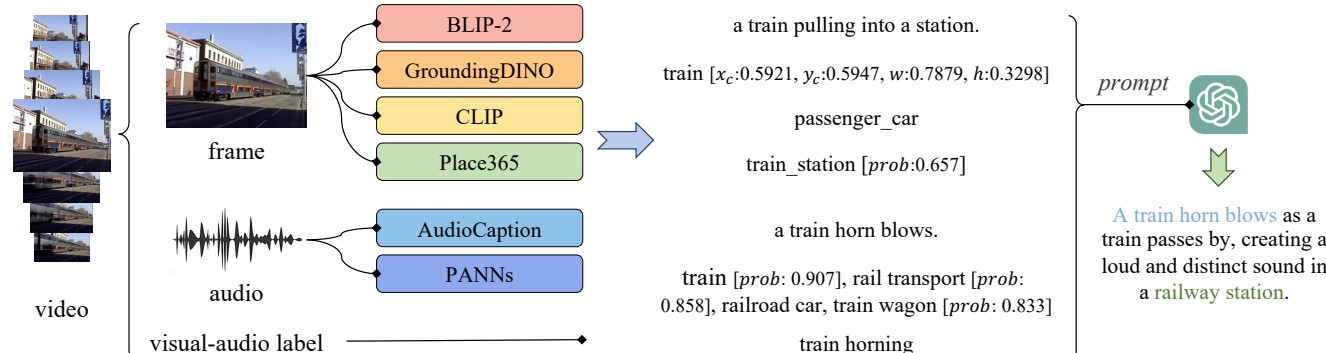

**Figure 3: Automatic pipeline for Auto-ACD collection. We utilize four open-source computer vision models to extract visual clues from the middle frame of videos, and two open-source audio understanding models to analyze the entirety of the audio content. Consequently, we combine the labels from the original dataset, and leverage Large Language Models (LLMs) to interpret and paraphrase these components into the final description.**

## 2 RELATED WORK

### 2.1 Audio-visual Learning

Within in-the-wild videos, audio-visual events occur simultaneously, establishing a profound connection between sound and imagery. [1, 2, 20, 43] employ audio-visual self-supervised learning to leverage audio-visual correspondence for enhancing representation learning. Specifically, [13, 52, 56] learn audio-text representation based on such correspondence. Audio-visual localisation [5, 18, 37, 38, 51] concentrates on identifying the positions of visual sound sources within video. Audio-visual segmentation [11, 26, 29, 39, 58] aims at predicting pixel-wise segmentation masks of sounding objects in visual scenes precisely. Such studies have further demonstrated the intrinsic correlation between audio and visual events in in-the-wild videos, which inspires us to create an audio-language dataset anchored in visual information.

### 2.2 Audio-visual Dataset

Large-scale audio-visual datasets are crucial for effective audio and video understanding. Two datasets are often involved in audio-visual learning: AudioSet and VGGSound. AudioSet [12] is a large-scale audio-visual dataset with multiple audio events labelled for each audio clip. It contains over 2M 10-second audio clips. AudioSet is a manually annotated dataset, with the help of a well-structured hierarchical ontology consisting of 632 audio classes guided by literature and manual curation. VGGSound [6] comprises 200K 10-second videos for 309 audio classes. This dataset was collected and annotated through an automated pipeline, with each video assigned only one label. Due to the strong correlation between in-the-wild video and audio, in this paper, we extract audio from audio-visual datasets and generate corresponding audio captions with the assistance of audio-visual clues.

### 2.3 Audio-language Dataset

Audio-language tasks, including audio-text retrieval, audio captioning, audio question answering and text-guided audio generation,

have greatly benefited from the availability of two widely-used audio captioning datasets: AudioCaps and Clotho. AudioCaps [21], a subset of AudioSet, consists of 50K 10-second-long audio clips, each with a single caption annotated. The annotators were provided with AudioSet tags as hints and videos if necessary. Clotho [10], on the other hand, comprises 6K audio clips lasting between 15 to 30 seconds, each with five captions annotated through a three-step process involving captioning, grammar correction, and rating by human annotators. However, due to the human annotation process, these datasets are limited in size, expensive and time-consuming. LAION-Audio-630K [53] acquires audio and descriptions from online foley websites, including popular platforms like Freesound[1] and BBC Sound Effects[2]. WavCaps [35] utilizes ChatGPT to filter and paraphrase these raw descriptions, resulting in a dataset of 400K audio-text pairs with cleaned text data resembling human annotations. The sentence is mostly simple since there is often only one sound event in an audio clip. As a result, models trained on these datasets could only learn the category of sound while neglecting rich information like the environment. To enhance the comprehension capabilities of the audio-text model, we need a more diverse set of textual and audio data.

### 2.4 Audio-language Learning

In the evolving landscape of AI research, the application of visual-language models in the audio-language arena marks a significant leap forward. Notably, [53] have adapted the CLIP model for audio-language contrastive learning, setting a precedent for innovative cross-modal research. This pioneering work exemplifies the growing interest in leveraging the success of visual-language models to enhance audio-language understanding. The audio-language domain is currently experiencing an explosion of interest across a variety of tasks. Researchers are not merely focusing on extracting semantic information from audio through tasks such as audio classification [17, 44], automatic audio captioning [34, 54], and audio

---

[1]https://freesound.org/
[2]https://sound-effects.bbcrewind.co.uk/

question answering [23, 28]. They are also venturing into more nuanced aspects of auditory perception, including the exploration of temporal dynamics in sound through audio event detection [3, 25]. This broadening scope encompasses additional auditory attributes such as counting sounds within scenes [41] and classifying environments based on their acoustic characteristics[9]. Undoubtedly, it is paramount to construct a comprehensive, large-scale, high-quality and information-rich audio-language dataset.

## 3 DATASET CONSTRUCTION

To develop a large-scale, audio dataset with rich language descriptions, we base on the assumption that visual scene understanding serves as a strong prior. For instance, synchronized videos frequently showcase auditory cues, and visual information serves as a precise representation of the acoustic environment in which the sound happens.

In an audio caption, it is desirable to incorporate sound attributes, location, and fine-grained labels. To achieve this, we can leverage publicly available tools or APIs to gather the necessary information for audio description and mutually verify the results. For instance, we can employ an object detection model to identify potential sources of sound, and an environmental classification model to extract scene categories. By extracting a wealth of information, we ensure the maximum coverage of accurate details, providing the language model with ample references.

### 3.1 Tools or APIs

Given one sample from existing large-scale video datasets, for example, AudioSet, VGGSound [6, 12], *i.e.*, denoted as $\mathcal{V} = \{f; a; y\}$, where $f$, $a$ and $y$ correspond to frame sequence, audio stream, and visual or audio labels, respectively. Our goal is to adopt a range of publicly available tools or APIs across the general AI community, *i.e.*, using off-the-shelf vision, language and audio models to construct language descriptions for audios, as shown in Fig. 3. In this section, we describe these tools in detail.

*3.1.1 Image Captioning.* We employ the off-the-shelf BLIP-2 [24] model, which obtains competitive results for image captioning. This tool has the ability to generate captions that encompass the entire image and accurately depict the primary subject or environment. In our case, we input the middle frame of the video into this model.

*3.1.2 Object Detection.* We use the pre-trained Grounding DINO model [30], to identify objects within the middle frame, and preserve all the detected entities along with their corresponding prediction confidence scores to ensure a comprehensive analysis.

*3.1.3 Image Labeling.* We adopt the pre-trained OpenAI CLIP [45] model for image classification. In this application, we utilize the prompt: "a photo of a {label}" to generate textual embedding, leveraging the category ontology from ImageNet [8].

*3.1.4 Place Recognition.* We employ the pre-trained PlaceCNN [57], to infer the environment context captured in videos. Given the robust correspondence between audio and visual signals, the environment depicted in the video is highly likely to represent the acoustic ambience in which the sound occurs.

---

| Prompting ChatGPT to generate caption for audio |
| --- |

I will give you some information from a video and an audio, this audio is separated from the video.

There is a caption for an audio, simple audio caption, this sentence simply describe what happens in the audio.

There are some audio tags: multiple audio tags, they indicate the audio events in this audio. number indicates the probability.

The audio-visual label is **dataset visual-audio label**.

I extract a key frame from one video, and this is the image caption of this frame: image caption; this is the image label: image label; this is the object detection: object detection; this is the place detection: place label.

Now, please help me write one audio caption using common vocabulary and no more than 24 words, providing a description of what happened in the audio, and infer where the audio happened. You can refer the above information, and some visual information is inaccurate and can be ignored. please using the audio-visual label check the audio event in your caption.

The sentence you write need to be like these following examples:
A bell chimes thrice as birds chirp in the background in the forest.
A lawnmower engine buzzing and stopping to take a few breaks on the lawn.
A machine being operated intermittently and people talking in the background in a factory.

**Figure 4: Detailed prompt provided to ChatGPT. For visualisation purposes, we use different colors to highlight diverse visual-audio cues.**

*3.1.5 Audio Tagging.* We use the pre-trained PANNs [22] to predict the tags of sounds within the audio, and preserve the top three predictions with their confidence scores. This represents a crucial source of auditory temporal information, particularly for sounds emanating from entities not visible within the frame.

*3.1.6 Audio Captioning.* We use the existing AudioCaption [55] model, to generate concise and brief captions. These captions resemble the style of AudioCaps, focusing solely on the categorical information of audio events, devoid of any additional descriptive attributes about the sound.

*3.1.7 Audio-visual Synchonisation.* We employ the pre-trained Synchformer [19] to conduct synchronization detection between video and audio. This process could filter out samples consisting of irrelevant or unsynchronized video and audio content. In this case, we input both video and audio respectively into this model for analysis.

*3.1.8 Existing Audio-Visual Labels.* In addition to the predictions from models, we also incorporate the provided labels of existing datasets into our pipeline. For instance, VGGSound [6] gives a single label for each video, while AudioSet [12] provides multiple labels. These labels serve in the original dataset, offering accurate yet incomplete audio-visual information.

*3.1.9 Summary.* As for the language model, we use the OpenAI ChatGPT[3], which demonstrates formidable capabilities in reasoning and inductive summarization, to assemble the above-mentioned descriptions or labels into comprehensive descriptions for audio. Many works, like BLIP-2[24], show that utilizing existing tools

---

[3]https://openai.com/chatgpt

**Table 1: The results of generated captions in Auto-ACD, with accurate content and ample surrounding information. Green and Yellow refer to "where" and "how" the audio sounds like.**

| No. | Generated Caption |
|-----|-------------------|
| 1. | Loud pops and bangs resonate as timbales are being played, creating rhythmic music in a room. |
| 2. | Water gurgles and bubbles as a boat glides through, creating a soothing and peaceful underwater ambience. |
| 3. | A woman speaks softly amidst the soothing sound of birds chirping, creating a serene atmosphere in a garden. |
| 4. | A motorcycle engine idles before revving up, creating a loud sound in an urban environment. |

appropriately can significantly enhance model performance. By leveraging audio-visual correspondence and the profound understanding capabilities of LLM, we generate precise audio captioning from the rich multi-modality clues acquired. In this case, we feed in a special prompt as shown in Section 3.2.

### 3.2 Caption Generation

Based on the visual and acoustic clues present in the video, we craft a structured language paragraph, and use it to prompt ChatGPT to generate descriptions for audio. As illustrated in Fig. 4, the process begins with formulating the specific task and criteria for the desired outcome, followed by inputting seven distinctive audio-visual cues into the prompt, accompanied by their corresponding confidence score. Additionally, we provide three sentence examples from AudioCaps or Clotho as instruction. For visualisation purposes, we here use a colour-coded system to distinguish various cues.

While generating captions, we explicitly ask ChatGPT to remove information that is inaudible, *i.e.*, illogical and visually oriented elements, for example, colours. As a result, the large language model is able to analyze the scenario from all provided clues, and generate language description for audio, with sound category, and environment. The generated caption results are shown in Table. 1.

### 3.3 Dataset Filtering

AudioSet is vast and diverse, while heavily marred by noise in many instances, for instance, gameplay live streams and explanatory videos. Conversely, VGGSound significantly emphasises the robust correlation between video and audio within the automated collection pipeline, thus requiring no further processing. As shown in Figure. 5, we formulate filtering criteria grounded in both the video-audio correspondence and the original labels. For each filter criterion, we conduct numerous trials followed by a manual verification, each filtering criterion achieves an accuracy rate exceeding 90%, resulting in the removal of 0.4 million videos in total.

*3.3.1 Raw labels.* AudioSet contains a plethora of explanatory videos with background music, wherein the visual and auditory information often do not correspond. Therefore, we eliminate videos from the multi-labels that encompass both speech and music.

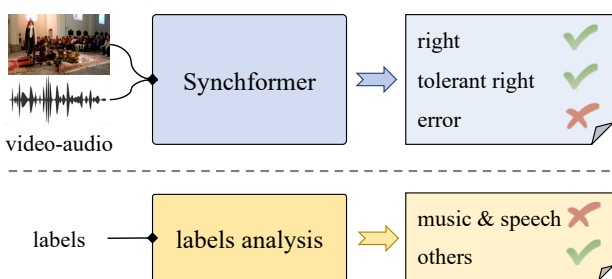

**Figure 5: Filtering process for AudioSet. We filter the dataset by assessing whether the video and audio are synchronized and analyzing the labels in the original dataset.**

*3.3.2 Audio-visual synchronisation.* To obviate the possibility of fortuitous inference errors, we subject each video to five synchronization evaluations, featuring random variations in start time and offset, with a tolerance threshold established at 0.6 seconds. Synchformer[19] employs a 0.2s offset to ascertain the precise audio-visual synchronization, whereas we utilize a broader offset to determine the audio-visual correspondence. The outcomes are categorized as follows: (1) Predictions aligning with the ground truth are deemed "correct"; (2) Predictions that diverge from the ground truth while with a discrepancy within 0.6 seconds are designated as "tolerant right"; (3) All other results are termed "error". To preserve as much accurate data as possible, videos classified as "error" in all five tests are removed from the dataset.

### 3.4 Dataset Statistics

As depicted in Fig. 2, we collect 1.5 million audio-language pairs from AudioSet and VGGSound in total. To the best of our knowledge, Auto-ACD is the first million-level audio-language dataset to date, with train, validation and manually filtered test sets. Auto-ACD surpasses the other datasets in terms of data volume, average sentence length, and contains a relatively wide verbal vocabulary. LAION-Audio-630K[53] sources from user uploads, contains a plethora of noisy details, for instance, device and timestamps, and features an exceptionally extensive vocabulary. Additionally, Auto-ACD stands as the only audio-language dataset that encompasses environmental information, not only delineates the type and source of sounds but also specifies the location of their occurrence, increasing the richness of contextual details.

In Table. 2, we present present a comparative analysis of captions from LAION-Audio-630K, WavCaps, and Auto-ACD for the same audio sample. Specifically, LAION-Audio-630K employs a keyword-to-caption model to transform the tag labels into captions. WavCaps utilizes ChatGPT to rephrase the tag labels into simple captions. It can be observed that captions in LAION-Audio-630K and WavCaps are concise and contain minimal information beyond the audio tags. In particular, LAION-Audio-630K may include sentences that deviate from common sense, for example, describing "rapping a tree" for an audio tag of "rapping". WavCaps, on the other hand, exhibits a monotonous sentence structure, such as "... sound can be heard". In contrast, Auto-ACD features longer sentences that provide a richer depiction of the audio scenes.

**Table 2: Caption comparison with LAION-Audio-630K and WavCaps, "LA.", "WavC." and "ACD" refer to LAION-Audio-630K, WavCaps and Auto-ACD, respectively.**

| No. | Dataset | Generated Caption |
|---|---|---|
| 1. | LA. | A person is rapping a tree. |
| | WavC. | Music plays with a man rapping. |
| | ACD. | A woman sings while hip-hop music plays in the background, creating a rapping audio event in a computer room. |
| 2. | LA. | a slushy water lily. |
| | WavC. | Stream noise, crowd and splashing sounds. |
| | ACD. | A crowd of people yells and cheers as water sloshes in the background at a water park. |
| 3. | LA. | a truck with a siren and a fire engine in an emergency. |
| | WavC. | A fire engine siren is heard. |
| | ACD. | An emergency vehicle siren blares loudly as a fire truck rushes through a residential neighbourhood. |
| 4. | LA. | a vehicle with a medium frequency of engine idling. |
| | WavC. | A medium engine sound can be heard. |
| | ACD. | A medium-sized engine is idling and vibrating, while an adult male speaks in the background near a running vehicle. |

## 4 ARCHITECTURE

We construct architectures targeting two general audio-language tasks, audio-language contrastive pre-training and automatic audio captioning, to further validate the effectiveness of Auto-ACD. In Section 4.1, we provide a detailed exposition of the architecture for audio-language contrastive learning. Further in Section 4.2, we introduce the framework for lightweight automatic audio captioning along with its loss function.

### 4.1 Audio-Language Constrastive Pre-training

To validate the efficacy of our proposed dataset, we train an audio-language model with standard contrastive learning, *e.g.*, infoNCE [45] loss, as shown in Fig.6. Specifically, we employ the pre-trained HT-SAT [7] as the audio encoder, and the pre-trained RoBERTa [32] as the language encoder. Both encoders were initialised from the pre-trained CLAP model [53], and further finetuned on our dataset. We term our final model as Audio-Text Retrieval (ATR).

Given an audio-text pair $(a^i, t^i)$, we utilise audio encoder $\mathcal{A}_{\text{enc}}$ and text encoder $\mathcal{T}_{\text{enc}}$ to extract audio embedding $e_a^i$ and text embedding $e_t^i$, respectively:

$$e_a^i = \mathcal{A}_{\text{enc}}(a^i), e_t^i = \mathcal{T}_{\text{enc}}(t^i)$$

The model is then trained with contrastive loss, wherein the paired audio and language embeddings are treated as positive, and unpaired ones as negative, with the following loss function:

$$\mathcal{L} = \frac{1}{2N} \sum_{i=1}^{N} (\log \frac{\exp \frac{e_a^i \cdot e_t^i}{\tau}}{\sum_{j=1}^{N} \exp \frac{e_a^i \cdot e_t^j}{\tau}} + \log \frac{\exp \frac{e_t^i \cdot e_a^i}{\tau}}{\sum_{j=1}^{N} \exp \frac{e_t^i \cdot e_a^j}{\tau}})$$

where $\tau$ represents the learnable temperature parameters.

During the training phase, we introduced word-level text masking. Before feeding sentences into the retrieval network, we randomly mask words within the sentences.

### 4.2 Automatic Audio Captioning

To demonstrate the effectiveness of our pre-trained audio backbone, we also use audio captioning for evaluation. Inspired by Clip-Cap [40] and AutoAD [14], we adopt a lightweight audio captioning model, where both the audio backbone and language model (GPT-2) are fixed, and only a mapping network is trained, as shown in Fig. 6.

Given an audio-text pair $(a^i, c^i)$, we use the pre-trained audio encoder to extract audio features $e_a^i = \mathcal{A}_{\text{enc}}(a^i)$, and we convert the caption into a token sequence, $c_1^i, \ldots, c_k^i$, where $k$ indicates the maximal length of text. Then, we design a mapping network $f_{\text{map}}$ to transform the extracted embedding into a set of prefix embeddings:

$$\mathcal{P}^i = f_{\text{map}}(e_a^i).$$

Like ClipCap and AutoAD, we take the prefix embedding set as the condition for predicting the next token in an auto-regressive language model. Therefore, during training, we minimize the negative log-likelihood of predicting the correct word:

$$\mathcal{L} = - \sum_{i=1}^{N} \sum_{j=1}^{\ell} \log p_\theta \left( c_j^i \mid \mathcal{P}^i, c_1^i, \ldots, c_{j-1}^i \right)$$

where $\theta$ represents the trainable parameters.

## 5 EXPERIMENTS

In this section, we evaluate three tasks, namely, audio-language retrieval, audio captioning and zero-shot classification.

### 5.1 Audio-language Retrieval

*5.1.1 Dataset.* We conduct audio-text retrieval experiments across several datasets: AudioCaps, Clotho, Auto-ACD$_{\text{VS}}$, and Auto-ACD. The distributions for the train, validation, and test sets in Audio-Caps, Clotho, and Auto-ACD are 50K/495/975, 3.8K/1045/1045, and 1.5M/2K/1K data pairs, respectively. Auto-ACD$_{\text{VS}}$, a subset of Auto-ACD, contains 190K data pairs exclusively sourced from VGGSound. Notably, in the case of Clotho, validation and test set in AudioCaps, each data pair consists of one audio sample accompanied by five corresponding captions, while the remaining data pairs only comprise

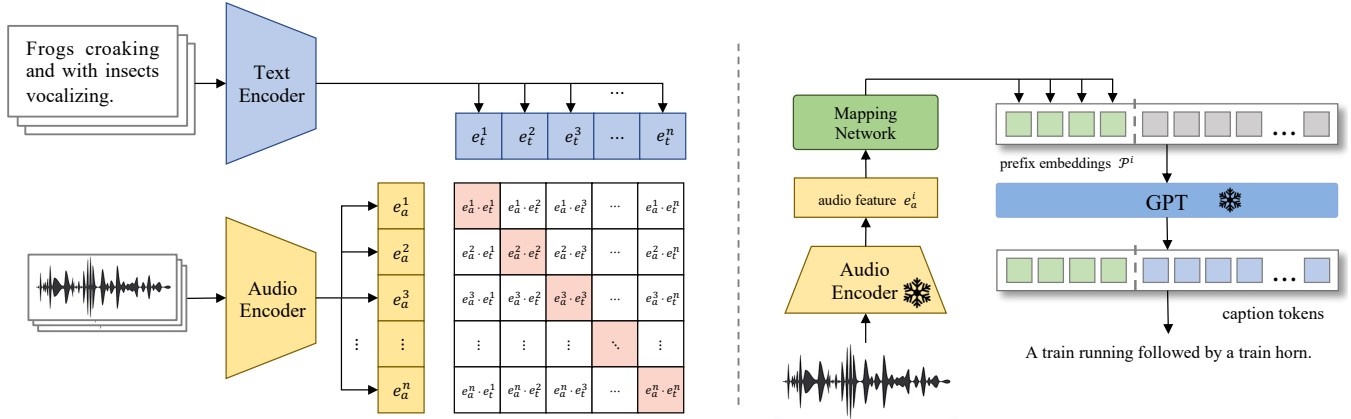

**Figure 6: Audio-language retrieval model and automatic audio captioning model frameworks. Similar to CLIP, the audio-language retrieval model consists of an audio encoder, text encoder, and contrastive loss. The automatic audio captioning model comprises a frozen audio encoder and language model, and a trainable mapping network.**

**Table 3: The audio-text retrieval results on AudioCaps, Clotho and ACD test sets. "basic", "LA." "Wav." and "ACD" refer to the combination of AudioCaps and Clotho, LAION-Audio-630K, WavCaps and Auto-ACD, respectively. "ACD$_{VS}$" is a subset of Auto-ACD, curated from VGGSound. " * FT" refers to fine-tuning the model on the target dataset.**

| Train Set | Model | AudioCaps Eval. | | | | Clotho Eval. | | | | Auto-ACD Eval. | | | |
|---|---|---|---|---|---|---|---|---|---|---|---|---|---|
| | | Audio→Text | | Text→Audio | | Audio→Text | | Text→Audio | | Audio→Text | | Text→Audio | |
| | | R@1 | R@10 | R@1 | R@10 | R@1 | R@10 | R@1 | R@10 | R@1 | R@10 | R@1 | R@10 |
| basic+LA.[53] | HTSAT-RoBERTa | 45.0 | 88.0 | 36.2 | 82.5 | 24.2 | 66.9 | 17.2 | 55.4 | 20.0 | 65.0 | 17.9 | 59.7 |
| basic+Wav.[35] | HTSAT-BERT | 51.7 | 90.6 | 39.7 | 86.1 | 23.4 | 63.4 | 19.5 | 58.2 | - | - | - | - |
| basic+ACD$_{VS}$ | HTSAT-RoBERTa | 50.5 | 90.6 | 39.8 | 86.9 | 24.2 | 62.9 | 20.0 | 58.9 | 39.2 | 86.2 | 39.6 | 85.7 |
| basic+ACD | HTSAT-RoBERTa | 53.7 | 91.7 | 39.5 | 85.4 | 17.7 | 52.6 | 15.3 | 52.1 | **47.1** | **91.2** | **49.0** | **92.3** |
| basic+ACD*FT | HTSAT-RoBERTa | **56.3** | **93.9** | **42.7** | **88.5** | **26.2** | **67.5** | **21.7** | **61.7** | - | - | - | - |

one audio-caption pair. It is worth mentioning that we manually filter and revise the Auto-ACD test set, to ensure the accuracy of the included information. During the annotation process, annotators review the original audio-visual data, amending or removing inaccuracies generated by the automatic pipeline, to guarantee the accuracy of the Auto-ACD test dataset.

*5.1.2 Auto-ACD Benchmark.* In addition to the Auto-ACD training set, we also randomly selected 2K data samples to form the validation set and 1K samples for the test set. We conduct a **manual** verification of the test set, by removing incorrect information from the language descriptions and rewriting inappropriate vocabulary expressions. This test set is used for evaluating both audio-language retrieval and automatic audio captioning tasks.

*5.1.3 Metrics.* In order to validate the rich and accurate information of our dataset, we compare the traditional metrics, Recall@$k$ performance, on commonly used datasets, for example, AudioCaps and Clotho. Simultaneously, we provide these metrics on the Auto-ACD test set, offering a comprehensive overview.

*5.1.4 Training Details.* We train our proposed Audio-Text Retrieval (ATR) model for 20 epochs, employing a batch size of 768, and utilizing the Adam optimizer with a warm-up phase, and an initial learning rate of 1e-4 with a cosine learning rate decay schedule. We use the same hyperparameters as those in the existing CLAP model configuration. The dimensions of both the audio encoder and text encoder output are 512. Additionally, we introduce 25% random masking on words in the sentences and randomly apply augmentations such as Noise and Gain to 50% of audio samples to enhance the model training. We further fine-tune the model on specific datasets, for example, Clotho and AudioCaps, with an initial learning rate of 2e-5 for 15 epochs.

*5.1.5 Results.* As shown in Table.3, we can draw the following key observations: (i) training on our proposed Auto-ACD$_{VS}$ dataset leads to a significant improvement in Recall@$k$ metrics. (ii) training on Auto-ACD results in a remarkable performance gain. This improvement is particularly evident when evaluating the model on the test set of AudioCaps, as AudioCaps is a subset of AudioSet and shares a similar data distribution with Auto-ACD. Such fine-tuning

processes enable the model to acquire a more comprehensive understanding of both audio and text information, thus enhancing retrieval performance. (iii) on the Auto-ACD benchmark, characterized by a more diverse lexicon and abundant language description, training on Auto-ACD datasets significantly outperforms the model trained on Laion-Audio-630K.

## 5.2 Automatic Audio Captioning

*5.2.1 Dataset.* In addition to the datasets mentioned in Section 5.1, we also use the MACS dataset [33], which comprises 3.9K audio-text data pairs, with each audio accompanied by two to five captions and several audio tags. In total, we train the automatic audio captioning model utilizing a total of 58k data pairs from Clotho, AudioCaps and MACS, and evaluate on Clotho and Auto-ACD test set.

*5.2.2 Metrics.* In addition to conventional captioning metrics, for example, Meteor [4], RougeL [27], Spider [31], we incorporate SentenceBERT [48] as additional evaluation metrics, that not solely rely on lexical alignment, but rather prioritize the semantic resemblance and accuracy of the captions' content.

*5.2.3 Training Details.* We devise two mapping networks, MLP and transformer, and selectively fine-tune the parameters of GPT during the training process. We set the number of prefixes to be 8, each with a dimension of 512. We train this audio captioning model on the MACS [33], Clotho and AudioCaps for 15 epochs with a batch size of 128 and an initial learning rate of 5e-4. In this task, we compare the audio encoder from our ATR model and the pre-trained CLAP [53], by only training the mapping network of both models on the benchmark datasets, namely, Clotho, and Auto-ACD.

*5.2.4 Results.* As shown in Table. 4, we can draw two observations: (i) The automatic audio captioning model, with the audio encoder initialised from our pre-trained ATR model, shows improved performance across all evaluation metrics than baseline. (ii) There is a more pronounced outcome when evaluated on Auto-ACD: the baseline approach's performance oversees a sharp decrease in the test set of Auto-ACD. We conjecture this is because the baseline features extracted from the CLAP model lack detailed descriptions of environmental information. While captioning model based on our ATR model shows a significant performance improvement, and is able to infer where the sound occurs precisely. This observation signifies that Auto-ACD showcases an extensive lexicon, enabling the portrayal of a given audio using various sentence structures. On the other side, it illustrates that models trained on our dataset will deduce the context in which the sound emanates.

**Table 4: The automatic audio captioning results on Clotho and Auto-ACD test sets. "S-BERT" refers to SentenceBERT, "Env." refers to whether the predicted captions contain environmental information.**

| Eval Set | Audio Encoder | Meteor | RougeL | Spider | S-BERT | Env. |
|---|---|---|---|---|---|---|
| Clotho | CLAP | 15.5 | 34.9 | 20.6 | 46.0 | ✗ |
|  | Ours | 16.6 | 36.2 | 21.2 | 47.4 | ✗ |
| Auto-ACD | CLAP | 9.9 | 23.0 | 19.6 | 8.7 | ✗ |
|  | Ours | 21.3 | 37.9 | 56.7 | 10.1 | ✓ |

## 5.3 Zero-shot Classification

*5.3.1 Dataset.* Auto-ACD stands out for integrating its incorporation of environmental information within its text descriptions. Following the training on Auto-ACD, we conduct environmental classification in two distinct scenarios. One scenario involved utilizing the urban acoustic scene dataset [16], known as *DCASE 2020 Mobile*, previously utilized in the DCASE 2020 challenge. The second scenario involved a collection of samples from the AudioSet evaluation set, annotated with child classes of "Acoustic environment" within the AudioSet ontology, referred to as *AudioSet Env*. To prevent data leakage, here we exclusively utilize the model pre-trained on Auto-ACD$_{VS}$ for this experiment.

*5.3.2 Metrics.* We approach zero-shot environment classification as an audio-text retrieval experiment, employing a conventional paraphrasing template: "The sound in [environment label]." We utilize Recall@1 as the metric for evaluating the environment classification outcomes in this experiment.

*5.3.3 Results.* The experimental results, as illustrated in Table. 5, highlight the superior environmental recognition capability of ATR pre-trained on Auto-ACD in comparison to CLAP. Notably, on the AudioSet Env, our model significantly outperforms CLAP, even though we only utilize a subset of Auto-ACD, Auto-ACD$_{VS}$, for pre-training without any data leakage from AudioSet into our training dataset, further serving as a testament to the rich and accurate environmental information in Auto-ACD.

**Table 5: Zero-Shot Acoustic Environment Classification. "*" refers to pre-training model on Auto-ACD$_{VS}$.**

| Model | DCASE 2020 Mobile | AudioSet Env |
|---|---|---|
| CLAP | 32.2 | 19.5 |
| Ours | 36.5 | 39.5* |

## 6 CONCLUSION

In this paper, we present an automatic pipeline for audio caption generation, accompanied by a large-scale and comprehensive audio captioning dataset comprising 1.5M data pairs. Furthermore, we evaluate the performance of various audio-language models on our dataset to authenticate the effectiveness, and provide a manually verified test set along with a benchmark for audio-language tasks. These experimental findings unveil the wealth of information and precise descriptions inherent in our data, facilitating the models to learn more robust audio-language representations.

Owing to the fact that a portion of our dataset originates from VGGSound, procured through an automatic pipeline. The transformation from online videos to precise audio-language pairs has evolved into a thoroughly automated and replicable procedure. Consequently, the acquisition of an expanded corpus of audio-language datasets is now a straightforward endeavour. Furthermore, as open-source computer vision models and Large Language Models (LLMs) undergo continuous refinement and advancement, the capacity to extract more precise audio-visual indicators improves, subsequently enhancing the precision of inferences and the quality of paraphrasing the final audio captions.

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
