# OpenReview forum: "Auto-ACD: A Large-scale Dataset for Audio-Language  Representation Learning"
_acmmm.org/ACMMM/2024/Conference — MM2024 Poster_

### Official Review · Reviewer_NVmz · 2024-05-23

**Rating:** 4
**Confidence:** 3

**Summary:**

The paper mainly introduces a new large-scale audio-language dataset, Auto-ACD. The dataset consists of audio separated from videos and captions generated by an automatic pipeline. The authors adopted visual-language models to extract visual clues from image frames and audio understanding models to analyze the the audio content, resulting in descriptions not only depict the type of sound and its source, but also the auditory environment and location of its occurrence. They also validate the efficacy of the proposed dataset by three downstream audio understanding tasks with the Audio-Text Retrieval model and the audio encoder trained on the proposed dataset.

**Strengths:**

1. Multi-modality Automated Pipeline

The paper introduces an innovative automatic approach for dataset creation, leveraging multimodal inputs such as video frames and audio streams, resulting in high-quality audio captions with environmental information, offering a more comprehensive understanding of the context in which sounds occur.

2. Benchmarking

The authors establish a novel benchmark for audio-text tasks, providing a baseline for future research and allowing for more thorough evaluation of models' capabilities.

3. Dataset Filtering

The inclusion of a dataset filtering process and human adjustments in the proposed benchmark helps maintain high-quality standards, ensuring that the final dataset is reliable and useful for training and evaluation.

**Limitations:**

1. Lack of Evaluation on Rewriting Quality of LLM

Despite using advanced models to gain the visual and audio clues, the LLM rewriting process might not always be satisfying to successfully preserve the nuances of auditory captions. For example, it may fail to transfer the appropriate auditory-only information and neglect the visual-related information, also with a chance to add or change the original meanings.

While the paper mentions thorough detection in audio-visual synchronisation, it lacks the effectiveness verification on the LLM outputs quality, which could lead to inaccuracies.

2. Over-reliance on CV Models

The dataset relies heavily on existing pre-trained models for various tasks like object detection and audio tagging. The quality of the dataset is therefore dependent on the accuracy and biases of six models. It lacks evaluation on accuracies of these model captioning results.

Moreover, there is an ambiguous boundary between an Audio-Language dataset and a Visual-Language dataset after involving visual clues in audio captioning.

3. Scalability Concerns

While the dataset is large, the process of expanding it further may require significant computational resources, especially as the pre-trained models and APIs evolve.

4. Evaluation Limitations

The paper may not cover all possible evaluation metrics or tasks that could benefit from the dataset, potentially understating its full utility or limitations. It is still addicted to some traditional audio-language downstream tasks.

**Suitability:**

3

---

### Official Review · Reviewer_5ZL5 · 2024-05-23

**Rating:** 5
**Confidence:** 3

**Summary:**

The paper presents a dataset for training and validation of methods for multi-modal tasks, in particular audio and text, such as generation of image captions, object detection, or generation of audio tags. The dataset consists of a large number of audio and text pairs, which are generated from existing video datasets through a set of tools. The dataset is used to train a machine learning model. Experiments show the performance of the model in comparison to other approaches. Furthermore, existing models have been trained with the dataset and the comparative results are presented.

**Strengths:**

* The dataset contains much more samples than comparative datasets.
* Additionally, the textual descriptions in the dataset include information about the environment.
* The experiments show, that training of models including the created dataset improves the recall metrics compared to training with other datasets.
* The dataset is mostly created with tools from publications.
* The evaluation is based on common metrics.

**Limitations:**

* Not all methods used in the creation of the dataset are from peer-reviewed publications, in particular, the use of ChatGPT makes the dataset creation non-reproducible. However, the dataset is publicly available and, hence, can be used.
* I would like to have a quality assurance for the caption generation, in particular the prompt-based removal of information that is inaudible.

**Suitability:**

3

---

### Official Review · Reviewer_DaBd · 2024-05-24

**Rating:** 3
**Confidence:** 3

**Summary:**

The paper outlines the creation of a large-scale, high-quality, audio-language dataset named Auto-ACD, which contains 1.5 million audio-text pairs. This dataset was labeled using a series of pre-trained models and APIs for tasks such as audio-visual synchronization, image caption generation, object detection, and audio tagging. The authors then utilized this information to prompt ChatGPT to generate descriptions for audio. They subsequently used the dataset in conjunction with AudioCaps and Clotho to train models for audio-language retrieval, audio captioning, and zero-shot classification. This was done to demonstrate the dataset's effectiveness compared to other datasets like WavCaps. Finally, the authors proposed a new audio-text benchmark called the Auto-ACD Benchmark.

**Strengths:**

The dataset is large-scale and fully automatically labeled. At the very least, it could function as a weakly-supervised, large-scale dataset for audio-text pretraining.

The paper is well-written, with each step of the data pipeline clearly illustrated.

**Limitations:**

The novelty of this paper is limited. In the current era, generating synthetic data or automatic labels using ChatGPT in combination with other APIs is not a new concept. Compared to LLM synthetic data generation, where GPT can clearly serve as a very strong teacher due to its superior reasoning capabilities, the use of ChatGPT in audio-text data generation is debatable since it lacks multimodal support.

The main issue lies in the quality of the data. The authors' claim that the dataset is of high quality is not entirely convincing. The dataset pipeline involves a series of separate models/APIs, each of which could introduce errors or hallucinations in the process. The pipeline then feeds the information in text format, which could inevitably contain errors from some models. ChatGPT lacks the context of the actual audio and was not trained for multimodality. In this scenario, ChatGPT merely functions as a grammar corrector or text summarizer, propagating any errors from the previous pipeline and potentially adding its own hallucinations.

The paper describes the dataset filtering process from AudioSet, but it appears that there is no dataset filtering process after the automatic construction of the dataset, and simply treat the data as it for finetuning. The author may potentially refine the dataset now that more powerful multimodal LLM exisits.

Alternatively, one may want to use Auto-ACD as weakly supervised data for pretraining. it would be necessary for author to demonstrate its effectiveness against even larger scale simple audio tag data that has less hallucinations and point of failure.

**Suitability:**

3

---

### Official Review · Reviewer_XdbV · 2024-05-25

**Rating:** 3
**Confidence:** 4

**Summary:**

This paper proposes a new dataset called Auto-ACD. The labelling process is automated by utilizing data and pretrained experts in the video and audio domain to construct audio captions.

**Strengths:**

- The author has made full use of the video information with several visual experts for the automatic audio labelling process, which is a primary distinction from the previous audio-only labelling process.
- The proposed dataset has shown improved results on both the Audiocaps and Clotho datasets on the retrieval task.
- The author has proposed a new evaluation benchmark and validated it manually.

**Limitations:**

I'm open to raising the score if the author can adequately address the following main weaknesses.

Main weaknesses:

- Since the dataset is automatically curated with LLM and other pattern recognition models, it’s important to validate it to confirm its usability carefully. The author has made adequate comparisons on audio retrieval. However, I found the evaluation of audio captioning and zero-shot audio classification not convincing enough. For example, the zero-shot audio classification is only performed in a limited domain (acoustic environment classification). However, most retrieval work [1, 2] must be verified on a comprehensive range of tasks to validate the performance, such as instrument classification, keyword spotting, emotion recognition, speaker counting, etc. Considering this, I am not convinced that the audio retrieval and captioning results are supportive enough.

- The dataset could be noisy, considering the visual information incorporated.

- The author introduces substantial augmentation on text (25% drop) and audio (add noise) for audio retrieval training. This is not a usual setup for audio retrieval tasks, so it would be helpful if the author could include the result without this augmentation on Auto-ACD.


Minor issues:

- Line 576: Multiple “present”


[1] Elizalde, Benjamin, et al. "Clap learning audio concepts from natural language supervision." ICASSP 2023-2023 IEEE International Conference on Acoustics, Speech and Signal Processing (ICASSP). IEEE, 2023.

[2] Wu, Yusong, et al. "Large-scale contrastive language-audio pretraining with feature fusion and keyword-to-caption augmentation." ICASSP 2023-2023 IEEE International Conference on Acoustics, Speech and Signal Processing (ICASSP). IEEE, 2023.

**Suitability:**

3

---

### Meta-Review · Area_Chair_krT7 · 2024-07-04

**Recommendation:** Accept (Poster)
**Confidence:** 5

**Metareview:**

The paper introduces a large-scale, high-quality, audio-language dataset, named Auto-ACD, comprising over 1.5M audio-text pairs. To evaluate its validity, it trains widely used models on the new dataset and shows performance improvement on various downstream tasks.

Though the novelty of the paper is limited, it introduces a new dataset for audio-related tasks such as audio-language retrieval, audio captioning, and zero-shot classification. Therefore, it supports multimodality (text-audio). After considering the paper, the reviewer's comments, and the rebuttal I recommend 'accept (poster)' for the paper.

The reviewers highlight the following strengths and limitations:

Strengths:
1. The dataset is large-scale and fully automatically labeled. It could function as a weakly-supervised, large-scale dataset for audio-text pretraining.
2. The dataset contains much more samples than comparative datasets.
3. The paper is well-written, with each step of the data pipeline clearly illustrated.

Limitations:
1. The method novelty of this paper is limited.
2. Each API component has only around ~80% accuracy (The issue of hallucinations of open-source tools.)